**Data Availability Statement:** Relevant excerpts are included in the body of the manuscript. Full transcripts are not publicly available as they contain sensitive information, which, though de-identified,

# How early is too early? Challenges in ART initiation and engaging in HIV care under Treat All in Rwanda—A qualitative study

Jonathan Ross[1]*, Charles Ingabire[2], Francine Umwiza[2], Josephine Gasana[2], Athanase Munyaneza[2], Gad Murenzi[2], Sabin Nsanzimana[3], Eric Remera[3], Matthew J. Akiyama[1], Kathryn M. Anastos[1], Adebola Adedimeji[4]

1 Division of General Internal Medicine, Montefiore Medical Center, Albert Einstein College of Medicine, Bronx, New York, United States of America, 2 Clinical Education and Research Division, Rwanda Military Hospital, Kigali, Rwanda, 3 Institute for HIV Disease Prevention and Control, Rwanda Biomedical Center, Kigali, Rwanda, 4 Department of Epidemiology and Population Health, Albert Einstein College of Medicine, Bronx, New York, United States of America

* joross@montefiore.org

## Abstract

### Introduction

HIV treatment guidelines recommend that all people living with HIV (PLWH) initiate antiretroviral therapy (ART) as soon as possible after diagnosis (Treat All). As Treat All is more widely implemented, an increasing proportion of PLWH are likely to initiate ART when they are asymptomatic, and they may view the relative benefits and risks of ART differently than those initiating at more advanced disease stages. To date, patient perspectives of initiating care under Treat All in sub-Saharan Africa have not been well described.

### Methods

From September 2018 to March 2019, we conducted individual, semi-structured, qualitative interviews with 37 patients receiving HIV care in two health centers in Kigali, Rwanda. Data were analyzed using a mixed deductive and inductive thematic analysis approach to describe perceived barriers to, facilitators of and acceptability of initiating and adhering to ART rapidly under Treat All.

### Results

Of 37 participants, 27 were women and the median age was 31 years. Participants described feeling traumatized and overwhelmed by their HIV diagnosis, resulting in difficulty accepting their HIV status. Most were prescribed ART soon after diagnosis, yet fear of lifelong medication and severe side effects in the immediate period after initiating ART led to challenges adhering to therapy. Moreover, because many PLWH initiated ART while healthy, taking medications and attending appointments were visible signals of HIV status and highly stigmatizing. Nonetheless, many participants expressed enthusiasm for Treat All as a program that improved health as well as health equity.

include personal narratives that could result in identification of individuals, and because permission for public dissemination of raw data was not solicited from the Rwanda National Ethics Committee. De-identified data that support the findings of this study may be made available for researchers who meet the criteria for access to confidential data upon reasonable request. For data inquiries, please contact Ms. Valentine Ingabire, secretary of the Rwanda National Ethics Committee, at info@rnecrwanda.org.

**Funding:** This work was supported by the U.S. National Institute of Mental Health (K23 MH114752); by the U.S. National Institutes of Health's NIAID, NICHD, NCI, NIMH, and NIDA, as part of Central Africa IeDEA (U01 AI096299); and by the Einstein-Rockefeller-CUNY Center for AIDS Research (P30 AI124414), which is supported by the following NIH Co-Funding and Participating Institutes and Centers: NIAID, NCI, NICHD, NHBL, NIDA, NIMH, NIA, FIC, and OAR.

**Competing interests:** The authors have declared that no competing interests exist.

## Conclusion

For newly-diagnosed PLWH in Rwanda, initiating ART rapidly under Treat All presents logistical and emotional challenges despite the perceived benefits. Our findings suggest that optimizing early engagement in HIV care under Treat All requires early and ongoing intervention to reduce trauma and stigma, and promote both individual and community benefits of ART.

## Introduction

Current World Health Organization (WHO) guidelines recommend that all people living with HIV (PLWH) initiate antiretroviral therapy (ART) as soon as possible after diagnosis, an approach known as *Treat All* [1, 2]. These guidelines have been adopted worldwide, and in sub-Saharan Africa, most PLWH are now initiating ART soon after diagnosis [3]. Initiating ART soon after HIV diagnosis improves individual outcomes including reducing time to viral suppression and decreasing mortality [4–7], and has the potential for health system benefits including more streamlined ART initiation workflows and reducing loss to follow-up between diagnosis and initiation [8, 9].

Despite the benefits of early ART, there are challenges to initiating medications soon after diagnosis. Studies of women initiating ART rapidly under Option B+, the pre-Treat All guidance recommending lifelong ART to pregnant and postpartum women, identified barriers to rapid initiation including insufficient time to process the diagnosis, feeling too healthy to start medication, and desire for confirmatory testing [10–12]. As Treat All is more widely implemented, an increasing proportion of PLWH are likely to initiate ART when they are asymptomatic, and they may feel that the benefits of ART are outweighed by the potential drawbacks of engaging in HIV care. Indeed, an emerging literature from sub-Saharan Africa suggests that PLWH initiating ART rapidly after diagnosis may be more likely to be lost to care than those initiating later [13–15]. At the health systems level, implementation of Treat All may place strain on health facilities. The SHAPE-UTT study, a large, multi-country study examining implementation of Option B+ in sub-Saharan Africa highlighted greater burden on healthcare workers, need to ration limited resources, and one-size-fits-all approach to scaling up ART as factors placing strain on health systems and potentially leading to negative outcomes [16–18]. The potential for similar dynamics with widescale implementation remains a concern [19].

As Treat All guidelines are increasingly adopted globally, understanding the experiences of PLWH under this paradigm is critical to identify barriers to care and develop effective interventions to ensure programmatic success. To date, there has been a lack of research to understand patient perspectives about initiating and engaging in HIV care in Sub-Saharan Africa under Treat All. We therefore undertook this qualitative study to explore the experiences of PLWH initiating ART in Rwanda, with a focus on challenges related to elements of healthcare delivery that changed with implementation of Treat All.

## Methods

### Study overview

The study was undertaken as part of a larger, multi-part project studying implementation of Treat All in Rwanda. Authors on this study included investigators in the Einstein/Rwanda Research and Capacity Building Program, a collaboration of American and Rwandan

investigators. Rwanda, a Central African nation with an estimated HIV prevalence of 3.0% [20], implemented Treat All in July 2016, with all PLWH immediately eligible for ART. National guidelines recommend that HIV pre-test counseling should be focused and stream-lined, and that ART initiation should occur as soon as possible after diagnosis [21]. Earlier research we conducted demonstrated rapid uptake of Treat All by health centers in Rwanda with most patients initiating ART soon after diagnosis [22]. We decided to conduct the present study to better understand patients' experiences of receiving care under this new paradigm, in particular to learn whether the larger numbers of individuals accessing ART, recommendation to initiate ART within 7 days of diagnosis, and the related reduced amount of pre-ART counseling might place additional burdens on some patients.

We conducted individual, semi-structured interviews with PLWH at two health centers in Kigali, Rwanda. The study, approved by the Rwanda National Ethics Committee and the Institutional Review Board of the Albert Einstein College of Medicine, was conducted according to the principles expressed in the Declaration of Helsinki and is reported in accordance with Consolidated Criteria for Reporting Qualitative Research (COREQ) guidelines (S1 Checklist) [23]. Participants provided written informed consent prior to enrollment in the study.

## Setting and participants

We recruited a convenience sample of participants from two health centers that each provide routine HIV care and treatment to approximately 2,000 PLWH. Participants were recruited after referrals from health center staff. Inclusion criteria were: 1) ≥18 years; 2) living with HIV; 3) receiving/had received care from study health centers. Exclusion criteria were: 1) unable to communicate in Kinyarwanda, and 2) unable to provide informed consent. We purposefully recruited younger participants (ages 18–24), who have historically had poorer HIV outcomes, and those who missed appointments over the year prior to enrollment (i.e. patient files demonstrated ≥1 occasions on which they did not attend a scheduled appointment), aiming for a sample that was at least one-third younger participants and one-third persons who missed appointments. Participants were compensated 8,000 Rwandan francs ($8 USD) for their time, and transportation to and from the health center for the research interview.

## Data collection

An interview guide informed by the socioecological model (SEM) [24] was developed to elicit perceived barriers to, facilitators of and acceptability of initiating ART rapidly, adhering to ART, and attending health center appointments as they related to individual experiences, social networks, health care settings, and Rwandan society (Table 1; S1 Table). Prior to data collection, the interview guide was piloted with research staff not involved in the current study to ensure questions were comprehensible and interview length was appropriate After obtaining informed consent, semi-structured interviews lasting 60–90 minutes were conducted in Kinyarwanda (the language most commonly spoken in Rwanda) by a nurse (FU) and a social worker (JG), both trained in qualitative interviewing. Interviewers were not clinicians at the study health centers and had no prior or subsequent relationships with participants. Each interview was conducted in a private room at the health centers with only the participant, an interviewer and a note taker present; interviews were audio-recorded, transcribed in Kinyarwanda and translated to English. Transcripts were reviewed and compared to field notes to ensure that they reflected all content that arose during interviews. Interview guides were iteratively refined to clarify and further explore emerging themes relevant to implementation of Treat All.

**Table 1. Example questions from interview guide.**

| DOMAINS OF CARE | QUESTION | LEVEL OF SOCIOECOLOGIC MODEL |
|---|---|---|
| **Experiences of HIV diagnosis** | 1. Tell me about when you were first diagnosed with HIV | Individual |
| **Experiences initiating ART** | 2. Tell me about your experience first starting medications? How did you feel physically and emotionally after starting medications? | Individual |
| | 3. Were there certain problems or challenges you had that made it difficult to start medication? What were they? | Individual, social network, institutional |
| | 4. Were there certain things that made it easier to start medication? What were they? | Individual, social network, institutional |
| | 5. How did the environment at the clinic/health center affect the process of starting medications? | Institutional |
| | 6. Are there other things that the health center or government could do to make it easier to start taking medication? What are they? | Institutional, societal |
| **Experiences adhering to ART** | 7. Tell me about any difficulties you have experienced in continuing to regularly take medication for HIV | Individual, social network, institutional |
| | 8. Are there certain things that make it easier to stay on medications? What are they? | Individual, social network, institutional |
| | 9. Are there other things you think the health center or government could do to make it easier to stay on medications? What are they? | Institutional, societal |
| **Experiences adhering to appointments** | 10. Tell me about any difficulties you have had in continuing to regularly come to appointments at the health center for your HIV care | Individual, social network, institutional |
| | 11. Are there certain things that make it easier to stay in care at the health center? What are they? | Individual, social network, institutional |
| | 12. Are there other things you think the health center or government could do to make it easier to attend appointments? What are they? | Institutional, societal |
| **Perceptions of Treat All and of health care delivery** | 13. The government of Rwanda has decided that every person diagnosed with HIV should receive ART, and that they should start medications as quickly as possible after diagnosis. How do you feel about this? | Societal |
| | 14. Are there parts of your HIV care that you think are missing or insufficient? What are they? | Institutional |
| | 15. Are there parts of your HIV care that you think are unnecessary? What are they? | Institutional |

## Data analysis and validation

English-language transcripts were analyzed using a mixed deductive and inductive thematic analysis approach to describe key barriers to and facilitators of initiating and adhering to HIV care under Treat All. Three investigators (JR, CI, MA) developed the initial coding scheme using the SEM to categorize common themes that were present in three transcripts. The coding scheme was refined based on inductive analysis of emergent themes from transcripts as well as field notes related to elements of healthcare delivery that changed with the implementation of Treat All; discrepancies were discussed in detail and resolved by consensus. Using DeDoose software [25], the final coding scheme was independently applied to all 37 interviews by 4 coders (JR, CI, FU, JG), with each interview being coded by at least 2 investigators. The coding team regularly reviewed progress and resolved issues by consensus. These discussions also helped to establish that saturation had been reached. After all interviews were coded, excerpts were reviewed, examining themes within codes as well as between codes and using the constant comparative method to identify, refine and consolidate emergent themes. At the completion of data analysis, three investigators (CI, FU, JG) conducted a focus group, inviting 8 study participants to validate the findings and ensure that major emergent themes were consistent with participants' experiences. This group was purposefully selected to represent the variation among study participants in terms of gender, age, health center, and engagement in care; all 8 individuals agreed to participation. Investigators presented key themes that emerged

from the analysis and solicited impressions from participants, incorporating feedback from this focus group into the final analysis.

## Results

From September 2018–March 2019, we interviewed 37 participants, of whom 27 (73%) were women and 15 (40%) aged ≤24 years (median age 31). Overall, 7 participants (19%) were diagnosed before and 30 participants (81%) after Treat All was implemented in Rwanda in July 2016. Median time from ART eligibility to ART initiation was 2 months (interquartile range: 0–8 months) and median time from ART initiation to the interview date was 18 months (interquartile range: 11–26 months). Among participants, 15 (40%) had missed ≥1 appointment in the preceding year. Analysis revealed four major themes regarding initiation of ART and adherence to treatment: (1) traumatization by HIV diagnosis as a barrier to initiating ART; (2) contrasting opinions about initiating ART rapidly; (3) ART and HIV-related appointments as stigmatizing; and (4) high level of support for Treat All as a policy providing early ART to all PLWH.

### Trauma of HIV diagnosis as a barrier to initiating ART

The experience of being diagnosed with HIV was described by most participants as a devastating and overwhelming event. Many recounted shock, trauma and grief when they received a positive result, feelings that were often compounded by isolation and anticipated stigma from their social networks. One participant explained, "*When I arrived at home, I was invaded by grief, as you can understand. I felt like I didn't want anyone to speak with me because I felt that anyone who knew me knew that thing* [HIV-positive status]. *It was very hard for me, to the extent I thought that living was not necessary for me. I didn't need to live; I stopped working and spent two months thinking about it.*" (35-year old male).

Many participants described how they equated their HIV diagnosis with death, anticipating that the virus would soon end their lives. For some participants, the emotional crisis stemming from the diagnosis was severe enough that they contemplated suicide. For example, "*Immediately after they told it to me, I thought there was no more life. I thought that was the end. The healthcare provider told me that I will continue to live—but me, I thought that even though I would continue to live, I would not live more than 10 days. When I thought about how a person who died of that contamination looked like, I said that I would not die in that way with a skin rash, losing my hair, skinny with visible bones on the whole body. I thought that there was no more life. I thought to kill myself after arriving home from the health center*" (38-year old female).

Because of these overwhelming feelings, many had difficulty accepting their diagnosis and engaging in treatment immediately after diagnosis. One participant noted, "*The healthcare providers gave me an appointment date so that I could start medication like others, but I didn't respect the appointment because I was not mentally well and I had not accepted it. I spent 2 months without going to the health center*" (27-year old male). Similarly, another participant summed up the challenges of starting to engage in treatment while still processing their diagnosis, stating, "*I haven't had a thought to take medication yet. . .they gave me medication but I was not yet sure that I had* HIV" (22-year old female).

### Contrasting opinions about initiating ART rapidly

Participants reported mixed feelings about rapid ART initiation, describing both the challenges of and the potential benefits of starting medications soon after diagnosis. For many, major individual-level barriers to initiating ART rapidly included lack of accepting their

diagnosis, fear of lifelong medication and severe side effects at ART initiation. For example, one participant discussed how the prospect of taking daily ART for the rest of his life was overwhelming to him so early in the course of his treatment: "*What terrified me is taking medications every day till I die. I wondered how I could spend 50 years on medications. . . I asked healthcare providers if I could not take a short break but they told me that I had to take pills every day. That traumatized me*" (31-year old female).

Other participants described severe side effects in the immediate period after initiating ART. While these typically subsided after several days or weeks, many participants described these experiences as adding to the challenges of starting to take medications during the unsettling first days after diagnosis, and making it difficult to develop a habit of adherence. One participant recalled, "*I started medication but it was not going well. I took medication and I felt dizzy, felt bad, had nightmares, had many things really. I could take medication one day, and didn't take it the next day. I wished I could die so that it ends*" (49-year old male).

For some, feeling healthy was a barrier to initiating ART, as they had difficulty reconciling their new HIV status with a lack of any feeling of illness. One participant noted, "*They have to give you some time so that your brain can accept that you are sick and have to be treated, and have to take medications without skipping any day for a long time*" (31-year old female).

On the other hand, many participants recognized the health benefits of early ART initiation, including maintaining health, preventing transmission to others, and avoiding the potentially stigmatizing physical manifestations of advanced HIV. One participant recalled, "*I thought that nothing else could make me strong except HIV medication. I took medication so that it made me stronger*" (33-year old female). Another noted, "*The Treat All program is very good because all people who are contaminated with the virus can immediately start medication instead of being severely sick at home and consequently neighbors and others can know it*" (33-year old male).

Moreover, some felt that initiating ART rapidly made it easier for them to accept their HIV status and engage in care. Participants described how receiving medications immediately after diagnosis could limit the potential for doubt and anxiety to sabotage therapy. For example, one described, "*I think it is very good because if they test you and give you medications on the same day, you don't feel very anxious, as you should feel if you start after a long time. It is helpful and prevents you from thinking a lot. It helps to accept the status and commit to taking the medication instead of continuing to think a lot about it*" (35-year old female).

## ART and HIV-related appointments as stigmatizing

Many participants described anticipated stigma associated with engaging in HIV care, particularly early in the course of disease when they were still learning to accept the diagnosis. They noted that whereas the physical manifestations of advanced HIV were now less of a concern, as these were less likely when initiating ART early, the medication itself, as well as appointments at the health center, now had the potential to signal their status to others.

At the individual and social network levels, participants described ART pills as symbols of disease, and as a result they were afraid to be seen taking their medications. For example, one noted, "*When I started medication, I tried to avoid people we interacted with. Sometimes your friend could come to the house, get into the house by surprise and could see your medication. I started to take distance from the time I started medication. . .I started to be choosy about guests to avoid that people could know it* [HIV-positive status]" (35-year old female). Others noted how the side effects of the medications, rather than physical manifestations of HIV, could signal their disease status to others: "*I developed lipodystrophy; the fats were built in abdomen, I*

*lost fats in my face, my cheeks went inside and I said that it was caused by the medication and I was afraid that people were going to know that I was contaminated*" (24-year old female).

Attendance at appointments at the health center also led to anticipated stigma among participants, who described the fear of being seen by community members or acquaintances who might disclose their status to others. Because of these concerns, participants often skipped appointments early on in their treatment course. For example, one participant described the experience of her first follow-up appointment: "*When they told me that I would take medication, I felt ashamed because I thought that I would meet with people who knew me. The first time I came, when I saw a neighbor woman they said she was contaminated, I went back home*" (33-year old male).

Often, the long wait times at the health center compounded these feelings, as they added to concerns about exposure. A participant noted that after Treat All was implemented, "*We became many patients. . .many people who come looking for the same service. When there are many people seeking the same service, I prefer to go back home instead of sitting on the queue while there are even people who know me*" (24-year old male).

## High level of support for Treat All as a policy providing early ART to all PLWH

Despite many of the challenges participants identified with initiating ART rapidly and adhering to care, there was an overall high degree of support for Treat All as a policy that would improve life for PLWH in Rwanda. Many participants appreciated how early ART initiation had physical and emotional benefits, contrasting this with prior policies that did not provide access to therapy for all. "[Before], *many people's health conditions could become worse and some could die. But now, when you start medication, you get hope to live longer*" (35-year old female).

In addition to the potential individual gains from universal, early ART, participants also described the community and societal benefits of Treat All. In particular, they noted that early ART initiation and viral suppression could reduce risk of transmission to others. Participants expressed gratitude towards the government for increasing access to ART and in the process improving health equity. One participant stated, "*[Before], someone could be sick and not know that he is sick, not take medication, develop opportunistic infection, and continue to infect many others. The decision of giving HIV medication to every person diagnosed who is aware of his status helps you to protect yourself and protect others.*" (35-year old female). Another commented that. "*[Treat All] is something I am happy with because everyone is eligible. Poor or rich people, they all get the same treatment without any money*" (35-year old female).

## Discussion

In this study of Rwandan patients' experiences of initiating ART under Treat All, we found that for many participants, the trauma related to HIV diagnosis, the fear of lifelong medication, and stigma related to both ART and appointment attendance were major barriers to initiating and adhering to HIV care. While participants expressed an appreciation for the individual and community benefits of ART, our results suggest that not all PLWH will be ready to initiate ART rapidly, and that successful implementation of Treat All will require early, patient-centered intervention to reduce diagnosis-related trauma and HIV stigma.

Participants in this study struggled with the trauma related to their HIV diagnosis, difficulties in accepting it, and feeling intimidated at the prospect of lifelong ART. Many also reported experiencing substantial side effects in the first days and weeks after initiating therapy, likely reflecting either physiologic or psychosomatic adjustment to their medications. These results

are similar to prior studies of newly-diagnosed PLWH in sub-Saharan Africa describing individual-level feelings such as disbelief, shock, and fear for one's life immediately after diagnosis, as well as status denial, adverse medication side effects, and feeling too healthy for medications as barriers to initiating ART [26–29]. Our study adds to the literature by describing how these play out in a paradigm where the amount of time between diagnosis and ART initiation has been reduced, sometimes to a matter of hours, and patients are thus forced to try to process all of these experiences simultaneously. The observed results suggest that acknowledging the anxieties related to lifelong ART and providing relevant, ongoing counseling during the ART adjustment period can support adherence in patients initiating medication rapidly after diagnosis.

Our findings also indicate that addressing the trauma experienced by many PLWH resulting from their diagnosis is of high importance in facilitating successful initiation of and longer-term adherence to ART. Prior studies of PLWH underscore the high levels of pre-diagnosis trauma in this population, the distress experienced during an encounter in which a diagnosis is received and the degree to which receiving a diagnosis can be a destabilizing force in people's lives [30, 31]. Studies from oncologic settings, where patients are also confronted with life-threatening diagnosis, suggest that whether the experience of receiving the diagnosis transforms into a trauma depends on factors including prior experience with the disease, prognosis, treatment options, existing support systems, and how the diagnosis is communicated [32, 33]. Incorporating a trauma-informed approach during the peri-diagnosis period that addresses these factors could potentially facilitate successful initiation of and retention on ART. A trauma-informed care counseling strategy has demonstrated success in improving adherence among Rwanda youth living with HIV [34]; additional studies are needed to better elucidate the association between trauma and clinical HIV outcomes, and to determine whether trauma-informed care in the immediate post-diagnosis period could specifically benefit PLWH as they initiate ART.

Many participants in this study reported feelings of shame, stigma and fear in the early period after HIV diagnosis, consistent with many other studies of PLWH in sub-Saharan Africa and globally [35, 36]. In particular, participants expressed high levels of internalized stigma, for example using terms such as *kwandura* ("contaminated," or "made dirty") to describe their HIV status, and noted how anticipated stigma and fear of HIV status disclosure impacted their early engagement with HIV care. Many described hiding medications from other household members, skipping doses to avoid accidental status disclosure, and feeling stigmatized by being physically present in the health center. With implementation of Treat All in Rwanda and globally, a larger proportion of PLWH are initiating ART earlier in the course of their illness. In many cases, therefore, ART essentially renders their HIV status visible, rather than any physical manifestations of the disease. Similarly, while anticipated and experienced stigma in health centers has been widely described as a barrier to engagement in care [28, 37], the eligibility of all PLWH for ART has led to larger numbers of PLWH accessing ART programs, increasing the number of other patients present and the amount of time spent waiting. As greater numbers of PLWH are offered ART in the early, asymptomatic stages of HIV, there is therefore a risk that the negative aspects of taking medication or attending appointments may outweigh any perceived benefits of ART. Our findings suggest that incorporating stigma-reducing interventions at multiple levels could allow patients initiating ART early and rapidly to more successfully engage in care. Such approaches might include individual-level strategies such as strength-based, stigma-reduction sessions as part of pre- and post-test counseling and early on after ART initiation [38]. At an institutional level, structural changes such as integration of HIV services into general outpatient clinics [39], multi-month

ART prescriptions [40], and long-acting injectable ART [41] could allow PLWH to view engagement with care as less burdensome and stigmatizing.

At the social network as well as societal level, policies emphasizing the collective benefits of Treat All could increase buy-in of early ART and facilitate the transition from diagnosis to successful engagement in care. A study examining health care worker perspectives on Treat All suggested community-wide education on benefits of early ART to increase awareness would facilitate ART initiation [42]. Many participants in our study were aware of the benefits of initiating ART early with respect to their own and others' health and expressed a high degree of support for Treat All as a national policy. We are not aware of other studies that have described PLWH's perspectives on Treat All policies, and thus our findings provide an important and reassuring contribution to the field. Our findings also add to the emerging literature suggesting that emphasizing the social benefits of Treat All, including decreasing the risk of HIV acquisition in the population and the contribution to the overall health of society could successfully address some of the barriers to care for PLWH [43].

This study has several limitations. First, we interviewed a sample of PLWH in the capital of a country with a highly functional HIV service delivery system and with a lower HIV prevalence than in much of southern Africa, which may limit the generalizability of our findings. Second, our study was limited to patients in care, and thus could not capture the experiences of those who had dropped out of treatment. Finally, interviews took place in the health center, despite being conducted by research and not clinical staff, which may have resulted in social desirability bias.

In conclusion, we found that newly-diagnosed PLWH in Rwanda struggled with the challenges of initiating ART rapidly, including the trauma of diagnosis, the fears of lifelong therapy, and the stigma associated with engaging in care. Yet participants in this study were able to understand the benefits of early ART and expressed support for a national program designed to get as many PLWH on ART as rapidly as possible. These results suggest that not all PLWH will be ready to initiate ART rapidly, and that individual, patient-centered approaches to engaging in care are necessary. Continued, successful implementation of Treat All in Rwanda and beyond can benefit from early and ongoing adherence support aimed at reducing trauma and stigma, preventing involuntary disclosure, and promoting both individual and community benefits of ART.

## Supporting information

**S1 Checklist. COREQ 32-item checklist.**
(DOCX)

**S1 Table. Example questions from Kinyarwanda interview guide.**
(DOCX)

## Acknowledgments

We would like to thank the participants for their time and insight, as well as staff at Gikondo Health Center and Kicukiru Health Center in Kigali, Rwanda. We also thank Dana Watnick PhD and Elissa Faro PhD for their qualitative research capacity-building efforts.

## Author Contributions

**Conceptualization:** Jonathan Ross, Gad Murenzi, Kathryn M. Anastos, Adebola Adedimeji.

**Data curation:** Jonathan Ross, Charles Ingabire, Francine Umwiza, Josephine Gasana.

**Formal analysis:** Jonathan Ross, Charles Ingabire, Francine Umwiza, Josephine Gasana, Matthew J. Akiyama, Adebola Adedimeji.

**Funding acquisition:** Jonathan Ross, Kathryn M. Anastos.

**Investigation:** Jonathan Ross, Charles Ingabire, Francine Umwiza, Josephine Gasana, Gad Murenzi, Adebola Adedimeji.

**Methodology:** Jonathan Ross, Adebola Adedimeji.

**Project administration:** Jonathan Ross, Charles Ingabire, Athanase Munyaneza, Gad Murenzi.

**Resources:** Athanase Munyaneza.

**Supervision:** Jonathan Ross, Gad Murenzi, Kathryn M. Anastos, Adebola Adedimeji.

**Validation:** Jonathan Ross, Sabin Nsanzimana, Eric Remera, Kathryn M. Anastos.

**Writing – original draft:** Jonathan Ross.

**Writing – review & editing:** Jonathan Ross, Charles Ingabire, Francine Umwiza, Josephine Gasana, Athanase Munyaneza, Gad Murenzi, Sabin Nsanzimana, Eric Remera, Matthew J. Akiyama, Kathryn M. Anastos, Adebola Adedimeji.

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
