## [Decision Letter · Decision Letter 0]

4 Mar 2021

PONE-D-20-27888

How early is too early? Challenges in ART initiation and engaging in HIV care under Treat All in Rwanda – a qualitative study

PLOS ONE

Dear Dr. Ross,

Thank you for submitting your manuscript to PLOS ONE. After careful consideration, we feel that it has merit but does not fully meet PLOS ONE’s publication criteria as it currently stands. Therefore, we invite you to submit a revised version of the manuscript that addresses the points raised during the review process.

The reviewers presented a number of concerns regarding the methodology and presentation of the study. They require a number of revisions and/or clarifications about the inclusion criteria and methods of data collection used. They also presented suggestions for your discussion section. Their comments can be viewed in full, below. 

We look forward to receiving your revised manuscript.

Kind regards,

Natasha McDonald, PhD

Associate Editor

PLOS ONE

Journal Requirements:

2. If the questionnaire used in the study was used in a language other than English, please provide a copy as Supplemental Information to ensure that you have provided sufficient details that others could replicate the analyses.

3.Please include captions for your Supporting Information files at the end of your manuscript, and update any in-text citations to match accordingly. Please see our Supporting Information guidelines for more information: http://journals.plos.org/plosone/s/supporting-information.

4.We note that you have indicated that data from this study are available upon request. PLOS only allows data to be available upon request if there are legal or ethical restrictions on sharing data publicly. For information on unacceptable data access restrictions, please see http://journals.plos.org/plosone/s/data-availability#loc-unacceptable-data-access-restrictions.

5.We note that the grant information you provided in the ‘Funding Information’ and ‘Financial Disclosure’ sections do not match.

Reviewers' comments:

Reviewer's Responses to Questions

**Comments to the Author**

1. Is the manuscript technically sound, and do the data support the conclusions?

Reviewer #1: Yes

Reviewer #2: Yes

2. Has the statistical analysis been performed appropriately and rigorously? 

Reviewer #1: N/A

Reviewer #2: N/A

3. Have the authors made all data underlying the findings in their manuscript fully available?

Reviewer #1: Yes

Reviewer #2: No

4. Is the manuscript presented in an intelligible fashion and written in standard English?

Reviewer #1: Yes

Reviewer #2: Yes

5. Review Comments to the Author

Reviewer #1: This topic is relevant and important for the Fast-Track strategy to end the AIDS epidemic by 2030. Test and Treat has been widely adopted throughout sub-Saharan Africa, whereby all HIV-positive individuals initiate antiretroviral therapy (ART) immediately upon diagnosis and continue for life. Feeling healthy may exacerbate barriers to ART initiation. Strategies to reach healthy clients are needed. Hence, this study and similar studies can provide the necessary insight about the appropriate strategies to reach healthy clients.

Setting and participants

1. Inclusion criteria were: 1) ≥18 years; 2) living with HIV; 3) 110 receiving/had received care from study health centers. For how long have they received care from the study centers? Duration on ART? 1 month, 2months, 12 months??

2. What was the exclusion criteria?

3. Participants were compensated 8,000 Rwandan francs ($8 USD). Compensation for what? Is compensation for their time and contribution? Was compensation given as an incentive to overcome barriers? The reason(s) for compensation should be explicit.

Data Collection

1. The authors should briefly describe their relationship with participants: The relationship and extent of interaction between the researchers and their participants can influence participants’ responses. For transparency, the authors should identify and briefly state their assumptions and personal interests in the research topic.

2. Interview guides were iteratively refined to refined to clarify. (Please delete repeated words)

3. Was the interview guide pilot tested in this population? If No, why was it not pilot tested?

4. Data saturation was not considered in this study. Why was data saturation not considered during data collection?

5. Were field notes made during and/or after the interviews/focus group? If yes, were they reflected in the results section?

6. During the interviews, was anyone else present besides the participants and researchers? The authors should clarify.

Data analysis and validation

1. Three investigators (CI, FU, JG) conducted a focus group with 8 study participants. How were these 8 participants selected?

Discussion

1. Our findings also add to the emerging literature suggesting that emphasizing the social benefits of HIV treatment such as………….

Reviewer #2: This paper describes patient challenges and perspectives related to “treat all” policies in Rwanda. The findings of the paper are not necessarily novel. The shock of HIV diagnosis, anxiety about lifelong therapy, and anticipated stigma are all well established across multiple studies. However, this is a well written paper that makes a succinct argument for both the benefits and drawbacks of test-and-treat policies from patient perspectives. As such, I feel it makes a contribution to the literature.

INTRODUCTION

1. Consider the term “Treat All” vs. other terms (e.g., “test and treat”, “universal test and treat”, “rapid start”).

2. See the special issue in Global Public Health 16(2), which looks at the health system issue of “test and treat”. These papers may help to strengthen your introduction.

METHODS

1. Were individuals eligible if they had initiated ART under PMTCT guidelines?

RESULTS

1. Additional details are needed in describing the sample. Did all participants get diagnosed during the “Treat All” period? How many of the women had initiated ART under PMTCT? How long had participants been diagnosed? How much time elapsed for participants between HIV diagnosis and initiation of ART?

2. Define “missed > appointment”.

3. Delete “All interviews were conducted in Kinyarwanda”. (previously stated in methods)

4. Change “Traumatization by HIV diagnosis” to “Trauma of HIV diagnosis”

5. Lines 181-184 describe a participant who got diagnosed and then avoided care for 2 months before starting ART. As mentioned above, the “time elapsed” needs to be described for all participants.

6. Lines 203-206 describes a participant who reported severe side effects. Is your impression that the reported side effects were truly a result of the medication, or psycho-somatic symptoms of the anxiety of a new diagnosis? I feel like that may deserve some exploration in the discussion, as this might be another potential drawback of a rapid start model, and an area for additional counseling.

7. The word “contaminated” is often used in quotes. Consider the Kinyarwanda translation and whether the word “infected” would be a better translation.

DISCUSSION

1. The call for “trauma-informed care” related to the traumatic experience of an HIV diagnosis deserves more consideration. A trauma informed approach is an holistic approach that recognizes individuals’ broader life histories and how historical and on-going traumas impacts individuals’ health and engagement with the health system. In calling for trauma-informed care, this needs to go beyond the acknowledgement of the HIV diagnosis as a traumatic event. I would advise looking at the literature on diagnoses of other life-threatening illness (e.g., cancer) to see whether a trauma-informed care approach has been used specifically to process the traumatic event of a new diagnosis, and/or what other strategies exist to help individuals move beyond the initial shock and anxiety related to a new diagnosis.

6. PLOS authors have the option to publish the peer review history of their article (what does this mean?). If published, this will include your full peer review and any attached files.

Reviewer #1: No

Reviewer #2: No

---

## [Author Response · Author response to Decision Letter 0]

6 Apr 2021

EDITOR

We have made changes to the formatting of the manuscript to meet PLOS ONE's requirements.

2. If the questionnaire used in the study was used in a language other than English, please provide a copy as Supplemental Information to ensure that you have provided sufficient details that others could replicate the analyses.

We have uploaded a Kinyarwanda version of the interview guide as Supplemental Information.

3. Please include captions for your Supporting Information files at the end of your manuscript, and update any in-text citations to match accordingly. 

Supporting Information file captions were included after the References, as requested.

4. We note that you have indicated that data from this study are available upon request. PLOS only allows data to be available upon request if there are legal or ethical restrictions on sharing data publicly. 

We have elaborated on data availability in the attached Cover Letter, as requested.

We have noted in the cover letter that grant P30 AI124414 was not awarded directly to any of the study authors, rather, it supports the Einstein-Rockefeller-CUNY Center for AIDS Research.

REVIEWER #1

1. Inclusion criteria were: 1) ≥18 years; 2) living with HIV; 3) receiving/had received care from study health centers. For how long have they received care from the study centers? Duration on ART? 1 month, 2months, 12 months??

We agree with the reviewer that additional information on the sample is relevant for context, and we have now included data on HIV diagnosis and time on ART, though these were not part of the inclusion criteria. Among the 37 participants included in the study, 7 were diagnosed prior to and 30 after Treat All implementation; median time from ART eligibility to ART initiation was 2 months, and median time on ART prior to the interview was 18 months (Page 10, Paragraph 1).

2. What was the exclusion criteria?

We thank the reviewer for identifying this omission. We note in the text that exclusion criteria were inability to speak Kinyarwanda, and inability to provide informed consent (Page 6, Paragraph 3). 

3. Participants were compensated 8,000 Rwandan francs ($8 USD). Compensation for what? Is compensation for their time and contribution? Was compensation given as an incentive to overcome barriers? The reason(s) for compensation should be explicit.

Thank you for this comment. Participants were compensated for their time and for transportation to and from the health center for the research interview. We now state this in the manuscript (Page 6, Paragraph 3).

4. The authors should briefly describe their relationship with participants: The relationship and extent of interaction between the researchers and their participants can influence participants’ responses. For transparency, the authors should identify and briefly state their assumptions and personal interests in the research topic.

We now note in the methods that the study interviewers were not clinicians at the health centers and had no prior relationships with research participants, which we agree can provide better context for readers of the manuscript (Page 7, Paragraph 1). We also noted in the limitations that the interview setting may have resulted in social desirability bias (Page 19, Paragraph 2). 

Additionally, we appreciate the suggestion to provide additional information regarding the authors’ background, interests and assumptions on the research topic, which we have done in the Methods (Page 5, Paragraph 3).

5. Interview guides were iteratively refined to refined to clarify. (Please delete repeated words).

We thank the reviewer for noticing this error, which has been corrected (Page 7, Paragraph 1). 

6. Was the interview guide pilot tested in this population? If No, why was it not pilot tested?

Before starting data collection, we piloted the interview guide with Rwandan research staff not involved in the current study to ensure questions were comprehensible and interview length was appropriate, which we now note in the manuscript (Page 7, Paragraph 1). In line with best practices in qualitative research (Brod, et al, Qual Life Res 2009, throughout the data collection period, the guide was iteratively refined to clarify and further explore emerging themes relevant to implementation of Treat All. 

7. Data saturation was not considered in this study. Why was data saturation not considered during data collection?

Based on study aims and available resources, we initially planned for a sample of approximately 30 participants, considered a reasonable estimate for achieving saturation in a sample (Baker, et al, National Centre for Research Methods Review Paper, 2012). While the logistics of conducting the study (i.e. need to conduct the analysis in English, duration of time needed for interview translation, and the geographic separation of the study team) limited our capacity to assess data saturation during the data collection process, regular discussions during the coding and analysis established that data saturation had been reached, as we now note in the text (Page 9, Paragraph 1). 

8. Were field notes made during and/or after the interviews/focus group? If yes, were they reflected in the results section?

We thank the reviewer for this comment. Field notes were taken by one of the two research staff with the participants, while the other one conducted the interview. These notes were discussed during team meetings and incorporated into the analytic process, as is now noted in the methods (Page 7, Paragraph 1; Page 9, Paragraph 1). 

9. During the interviews, was anyone else present besides the participants and researchers? The authors should clarify.

We have indicated in the text that only research participants and interviewers were present during the interviews (Page 7, Paragraph 1). 

10. Three investigators (CI, FU, JG) conducted a focus group with 8 study participants. How were these 8 participants selected?

We have added text in the manuscript to describe how this group was purposefully selected to represent diverse viewpoints from the overall study sample in terms of gender, age, degree of clinical engagement (i.e. missed appointments); we note that all 8 participants who were invited to participate accepted (Page 9, Paragraph 1).

11. Our findings also add to the emerging literature suggesting that emphasizing the social benefits of HIV treatment such as…………. 

We have added to the text to provide examples of social benefits of HIV treatment such as an overall decreased risk of HIV acquisition in the population, a healthier population, etc. (Page 19, Paragraph 1). 

REVIEWER #2

1. Consider the term “Treat All” vs. other terms (e.g., “test and treat”, “universal test and treat”, “rapid start”). 

Although both “Universal Test and Treat” and “Treat All” are utilized widely in the literature, we have chosen to use the latter term for several reasons. This is a term used by the World Health Organization to refer to policies it recommended in 2016 to provide ART for all people living with HIV, and to do so as soon as possible after diagnosis, and thus encompasses both the concepts of “universal treatment” as well as “rapid start.” Additionally, the term “Treat All” is preferentially used by the Rwandan Ministry of Health for clinical guidelines, research manuscripts and other publications. 

2. See the special issue in Global Public Health 16(2), which looks at the health system issue of “test and treat”. These papers may help to strengthen your introduction. 

We thank the reviewer for this suggestion; these articles were not available at the time of initial submission. We have reviewed this literature and incorporated several suggested citations into the introduction, highlighting similarities between Option B+ and Treat All with respect to potential impacts of policy implementation (Page 4, Paragraph 2). 

3. Were individuals eligible if they had initiated ART under PMTCT guidelines?

As noted above (Reviewer 1, critique 1), eligibility was not limited to partcipants initiating ART under Treat All, and initial interviews included participants who had initiated ART under earlier eligibility guidelines. Because challenges initiating ART quickly after diagnosis emerged as a major theme, we thereafter focused recruitment towards participants who had first initiated ART after Treat All guideline implementation. Overall, 81% of the sample initiated ART after Treat All implementation, as noted in the manuscript (page 10, paragraph 1).

4. Additional details are needed in describing the sample. Did all participants get diagnosed during the “Treat All” period? How many of the women had initiated ART under PMTCT? How long had participants been diagnosed? How much time elapsed for participants between HIV diagnosis and initiation of ART?

Among all participants included in the study, median time to ART initiation from HIV diagnosis (for those initiating under Treat All) or ART eligibility (for those initiating prior to Treat All implementation) was 2 months; median time from ART initiation to the interview date was 18 months. This is now described in the text (page 10, paragraph 1). 

5. Define “missed appointment”.

In the study health centers, if patients do not attend a scheduled clinical appointment, nurses call them to attend the subsequent day. If this attempt is unsuccessful, the appointment is marked as “missed” in the patient file. Health center nurses participating in recruitment efforts reviewed the patient files to target recruitment towards these patients. We have clarified this in the text (Page 6, Paragraph 3).

6. Delete “All interviews were conducted in Kinyarwanda”. (previously stated in methods). 

We thank the reviewer for noting this redundancy, and have deleted the text accordingly (Page 10, Paragraph 1).

7. Change “Traumatization by HIV diagnosis” to “Trauma of HIV diagnosis.” 

We agree with the reviewer that this terminology is clearer and have made the suggested change (Page 10, Paragraph 2). 

8. Lines 181-184 describe a participant who got diagnosed and then avoided care for 2 months before starting ART. As mentioned above, the “time elapsed” needs to be described for all participants.

Thank you for this comment. As noted above, we have provided data on timing of HIV diagnosis and ART initiation, as well as duration on ART (Page 10, Paragraph 1). 

9. Lines 203-206 describes a participant who reported severe side effects. Is your impression that the reported side effects were truly a result of the medication, or psycho-somatic symptoms of the anxiety of a new diagnosis? I feel like that may deserve some exploration in the discussion, as this might be another potential drawback of a rapid start model, and an area for additional counseling.

Side effects at ART initiation, particularly nausea, fatigue and dizziness, were commonly reported among participants in this study. During the study period, the most common first-line ART medications in Rwanda were tenofovir, lamivudine, efavirenz and abacavir, which commonly can cause these types of side effects. It is difficult – both as investigators and as clinicians – to disentangle to what degree these adverse effects may be driven by physiologic versus psycho-somatic processes. However, the overall impact on initiating ART of either mechanism, or both together, can be a negative one. We agree with the reviewer that counseling can emphasize the potential for adverse physiologic and psychosomatic effects of ART and provide patients with reassurance that these effects are typically short-lived, and comment briefly on this in the Discussion (Page 16, Paragraph 2).

10. The word “contaminated” is often used in quotes. Consider the Kinyarwanda translation and whether the word “infected” would be a better translation.

Thank you for this comment. The Kinyarwanda word participants used was kwandura which is best translated as “contaminated” or “made dirty.” Because this was the word used by participants, we feel that it is most appropriate to retain it in the quotes presented in the manuscript. We comment briefly on the use of this word as a reflection of the high levels of internalized stigma among participants in the Discussion (Page 17, Paragraph 2).

11. The call for “trauma-informed care” related to the traumatic experience of an HIV diagnosis deserves more consideration. A trauma informed approach is an holistic approach that recognizes individuals’ broader life histories and how historical and on-going traumas impacts individuals’ health and engagement with the health system. In calling for trauma-informed care, this needs to go beyond the acknowledgement of the HIV diagnosis as a traumatic event. I would advise looking at the literature on diagnoses of other life-threatening illness (e.g., cancer) to see whether a trauma-informed care approach has been used specifically to process the traumatic event of a new diagnosis, and/or what other strategies exist to help individuals move beyond the initial shock and anxiety related to a new diagnosis.

We very much appreciate this suggestion. We agree with the reviewer that trauma – beyond that of HIV diagnosis – is highly prevalent among PLWH, and that trauma and HIV intersect in fundamental ways that impact health and engagement in health care. This is particularly true in Rwanda, given the collective trauma of the 1994 conflict and genocide. We have expanded on the possibility of how trauma-informed care can be applied in the discussion (Page 17, Paragraph 1).

---

## [Decision Letter · Decision Letter 1]

30 Apr 2021

How early is too early? Challenges in ART initiation and engaging in HIV care under Treat All in Rwanda – a qualitative study

PONE-D-20-27888R1

Dear Dr. Ross,

We’re pleased to inform you that your manuscript has been judged scientifically suitable for publication and will be formally accepted for publication once it meets all outstanding technical requirements.

Kind regards,

Matt A Price

Academic Editor

PLOS ONE

Additional Editor Comments (optional):

Reviewers' comments:

Reviewer's Responses to Questions

**Comments to the Author**

1. If the authors have adequately addressed your comments raised in a previous round of review and you feel that this manuscript is now acceptable for publication, you may indicate that here to bypass the “Comments to the Author” section, enter your conflict of interest statement in the “Confidential to Editor” section, and submit your "Accept" recommendation.

Reviewer #1: All comments have been addressed

2. Is the manuscript technically sound, and do the data support the conclusions?

Reviewer #1: Yes

3. Has the statistical analysis been performed appropriately and rigorously? 

Reviewer #1: N/A

4. Have the authors made all data underlying the findings in their manuscript fully available?

Reviewer #1: No

5. Is the manuscript presented in an intelligible fashion and written in standard English?

Reviewer #1: Yes

6. Review Comments to the Author

Reviewer #1: The authors have addressed the comments. I have no further comments to make regarding the manuscript.

7. PLOS authors have the option to publish the peer review history of their article (what does this mean?). If published, this will include your full peer review and any attached files.

Reviewer #1: **Yes: **Dr Olumuyiwa Omonaiye

---

## [Editor Report · Acceptance letter]

4 May 2021

PONE-D-20-27888R1 

How early is too early? Challenges in ART initiation and engaging in HIV care under Treat All in Rwanda – a qualitative study 

Dear Dr. Ross:

I'm pleased to inform you that your manuscript has been deemed suitable for publication in PLOS ONE. Congratulations! Your manuscript is now with our production department. 

Kind regards, 

on behalf of

Dr. Matt A Price 

Academic Editor

PLOS ONE